# Balance Assessment Using a Handheld Smartphone with Principal Component Analysis for Anatomical Calibration

**DOI:** 10.3390/s24175467

**Published:** 2024-08-23

**Authors:** Evan C. Anthony, Olivia K. Kam, Stephen M. Klisch, Scott J. Hazelwood, Britta Berg-Johansen

**Affiliations:** 1Mechanical Engineering Department, College of Engineering, California Polytechnic State University, San Luis Obispo, CA 93407, USA; ecanthon@calpoly.edu (E.C.A.); shazelwo@calpoly.edu (S.J.H.); 2Biomedical Engineering Department, College of Engineering, California Polytechnic State University, San Luis Obispo, CA 93407, USA; okkam@calpoly.edu (O.K.K.); bbergjoh@calpoly.edu (B.B.-J.)

**Keywords:** inertial measurement unit (IMU), anatomical calibration, accessibility, balance assessment, Principal Component Analysis, wearables, smartphone

## Abstract

Most balance assessment studies using inertial measurement units (IMUs) in smartphones use a body strap and assume the alignment of the smartphone with the anatomical axes. To replace the need for a body strap, we have used an anatomical alignment method that employs a calibration maneuver and Principal Component Analysis (PCA) so that the smartphone can be held by the user in a comfortable position. The objectives of this study were to determine if correlations existed between angular velocity scores derived from a handheld smartphone with PCA functional alignment vs. a smartphone placed in a strap with assumed alignment, and to analyze acceleration score differences across balance poses of increasing difficulty. The handheld and body strap smartphones exhibited moderately to strongly correlated angular velocity scores in the calibration maneuver (r = 0.487–0.983, *p* < 0.001). Additionally, the handheld smartphone with PCA functional calibration successfully detected significant variance between pose type scores for anteroposterior, mediolateral, and superoinferior acceleration data (*p* < 0.001).

## 1. Introduction

Due to the constant evolution of technology, the biomechanical field has been able to expand to more accessible and mobile motion assessment. The standard use of laboratory-based optical motion capture systems, force plate platforms, and other fixed lab equipment has begun shifting to the use of wearable sensors such as inertial measurement units (IMUs). These devices are an increasingly promising solution for balance assessment, also known as posturography [1]. Evaluating balance is a treatment response indicator for patients with neurological ailments like Parkinson’s disease, epilepsy, and multiple sclerosis [2]. Balance deficits and postural instability affect 39–62% of individuals with traumatic brain injury, leading to longer hospital stays, increased fall risk, delayed recovery, medical complications, gait abnormalities, and restrictions in daily and social activities, with these impairments often persisting as chronic issues despite initial improvements [3]. An estimated 30% of adults over 65 fall each year, with one in three never returning home after a hip fracture, but early intervention through accurate screening and remote assessment can reduce fall incidence by up to 50% [4]. These target groups who struggle with balance and mobility issues drive modern solutions to evolve into more accessible balance assessments.

The most common method of the clinical assessment of balance is the Berg Balance Scale, composed of a 14-item scale that quantitatively defines balance and risk for falls in the older community. This system requires a visit to the clinic. Additionally, participants often consistently achieve the maximum score, and changes within participants are only noticeable when balance is rapidly changing [5]. As previously mentioned, the laboratory gold-standard use of force plates uses center of pressure and ground reaction force vectors to measure balance, but this method involves many challenges, including its immobility, expensiveness, and the level of expertise required to operate the equipment and software [6]. Some solutions address the portability challenge of force plates, using portable force plates to evaluate postural sway in an at-home setting [7]. While this would eliminate the portability issue of force plates, this would still not be the most accessible method as this would require the purchase of additional dedicated equipment. IMUs, including the ones typically found in modern smartphones, typically consist of accelerometers, gyroscopes, and magnetometers, and offer the capability to assess gait and balance directly in real-world settings, eliminating the need for a dedicated laboratory. This wearable technology generates a time series of raw data that can be digitally translated to spatiotemporal parameters, as well as raw acceleration and angular velocity [8]. 

IMUs, particularly those found in smartphones, have been proven to be a reliable tool in analyzing static balance poses compared to force plates. Apple’s iPhones (Apple, Cupertino, CA, USA) offer robust capabilities to measure steps and exercise, with the use of an accelerometer to detect movement. iPhones are projected to remain one of the leading smartphones in the global market, with projected sales of USD 241 million in 2025 and USD 238 million in 2026 [9]. Smartphones equipped with IMUs offer a cost-effective and portable solution for balance assessment, with high ownership rates among all ages of adults. Despite their potential, few studies have validated smartphones against gold standard techniques [10].

Anatomical calibration is a process used in biomechanics and motion analysis to accurately align and relate the coordinates of a motion capture system or other measurement device with the anatomical landmarks and axes of the human body. Such calibration ensures that the data collected from the sensors accurately reflect the actual movements and positions of the body systems being studied [11]. However, most balance studies using smartphones have used a body strap and assumed alignment of the smartphone and anatomical body axes. The use of a body strap and assumed alignment may reduce accessibility in real-world settings and introduce errors (e.g., due to natural curvatures of the spine if the smartphone is placed on the lower back) that result in incorrect alignment to the global coordinate system [12]. For example, a recent study validating a smartphone app’s ability to evaluate both gait and balance, including static balance poses, against a 3D motion capture system utilized a sacroiliac belt to hold the smartphone at the participant’s lower back (around L5/S1) [13]. A recent review of studies using wearable sensors to evaluate standing balance found that 81% of studies also placed the device on the lower back [12]. The precise placement of these sensors is often challenging due to the irregular shape of the human limbs and anatomical variability from person to person, as well as the size constraints and adjustability of body straps. Additionally, with assumed alignment of the smartphone requiring the purchase of a well-fitting body strap, this may further reduce the accessibility of this balance assessment method. To address these limitations, in this study we used a functional alignment method, described and validated in our recent balance study using smartwatches [1], so that a smartphone can be held by the participant in a comfortable position. This method utilizes a single calibration maneuver and Principal Component Analysis (PCA) to align the smartphone data to the global anatomical body axes, i.e., the anteroposterior (AP), mediolateral (ML), and superoinferior axes (SI). Our recent study using the same PCA algorithm on an Apple Watch (Apple, Cupertino, CA, USA) found that the forward-flexion maneuver exhibited moderate to strong correlations with an IMU in a body strap, and was able to detect statistically significant differences between balance poses of increasing difficulty [1]. A notable conclusion of this prior study was a lack of significant differences in root mean square (RMS) angular velocity and acceleration-based balance scores between the least difficult pose (the double-leg stance) and the intermediate difficulty pose (the semi-tandem stance). For this reason, the tandem pose was selected for the current study as a more difficult pose. 

The objectives of this study were to (1) determine the strength of correlation between angular velocity scores from a handheld smartphone with PCA functional alignment vs. an assumed alignment smartphone placed on the lower back in a body strap during a forward-flexion calibration maneuver, and to (2) analyze within-participant acceleration score differences across three balance poses of increasing difficulty for both the handheld and body strap smartphones. 

## 2. Materials and Methods

The experimental protocols were approved by the Cal Poly Institutional Review Board for Human Subjects Research and were designed to minimize risk to human participants.

### 2.1. Participant Recruitment

In total, 22 participants (10 male, 12 female; aged 19–24) were recruited from the San Luis Obispo, CA community. Individuals with lower-extremity injuries or pain within the last 6 months were excluded from the study. No grouping of the participants was performed as this study is a comparison of measurement methods.

### 2.2. Balance Experiment

Balance experiments were composed of three 30 s static poses of increasing difficulty. Two iPhone 11 smartphones (Apple, Cupertino, CA, USA), one in a handheld position and one placed in a body strap, were used to record inertial acceleration, angular velocity, and orientation data. Data were recorded on each smartphone using the MATLAB App (MathWorks, Natick, MA, USA) at 100 Hz. The body strap smartphone was placed against the participant’s back with the phone’s Z-axis aligned anteriorly and Y-axis aligned superiorly (Figure 1). Two sizes of body straps—one larger and one smaller—were utilized to accommodate participants of varying sizes. Anatomical placement of the body strap smartphone was not standardized but it was placed at the participant’s thoracic spinal level on the lower back. 

The handheld smartphone was held in the participant’s hand against their chest in a crossed-arm position (Figure 2). Participants were instructed to hold the iPhone in whichever hand and wrist position they felt most comfortable so the smartphone would experience minimal motion artifact relative to their body. 

Informed consent and anthropometric measurements were obtained for each participant. Participants were shown how to perform each balance pose and encouraged to practice. Before each of the three 30 s balance poses, participants were fitted with both smartphones and performed a forward-flexion calibration maneuver. Participants were asked to perform the maneuver at a comfortable depth and speed. The forward-flexion maneuver (i.e., bending forward and back to standing about the hips) was used in a PCA-based algorithm [1] that can identify the orientation of the ML axis using angular velocity data taken during the maneuver. The forward-flexion calibration maneuver is a one-dimensional rotation about the participant’s ML axis (Figure 3).

Participants performed three 30 s balance poses (Figure 4). Participants who failed to hold the balance pose or the handheld smartphone were asked to reattempt the pose. The double-leg (DL) stance was performed in a neutral standing position with both feet on the floor, about shoulders’ width apart. The tandem (T) stance was performed with the dominant foot forward and non-dominant foot tucked behind. Finally, the single-leg (SL) stance was performed with the participant’s non-dominant leg on the floor and dominant leg bent at a 90-degree angle. Participants were asked to look at a marker on the wall ahead of them during each pose. At the end of the 30 s, participants were asked to perform a small jump to indicate the end of the pose (see Section 2.3). 

### 2.3. Data Acquisition and Post-Processing

Inertial acceleration, angular velocity, and orientation data were recorded at 100 Hz with the MATLAB Mobile App during both the calibration maneuver and balance pose, which followed in each experiment. Data for each participant were stored in the MathWorks Cloud (MathWorks, Natick, MA, USA) to be accessed via the desktop application for post-processing. The IMU sensor data from the iPhone were processed and optimized through sensor fusion and filtering. The iPhone is equipped with a LIS302DL 3-axis MEMS-based accelerometer that uses a low-pass filter to reduce high-frequency noise and smooth out the data. The MATLAB mobile app uses additional sensor fusion algorithms to estimate orientation and position over time, taking data from the accelerometer, gyroscope, and magnetometer of the iPhone [14,15]. The post-processing of data, described below, included gravity acceleration removal, trimming, filtering, and generating balance assessment scores. 

Data from both smartphones were imported into a custom MATLAB script. Gravity was removed from acceleration data by generating gravity vectors relative to the inertial coordinate system from the orientation data outputted from the MATLAB App. These gravity vectors were subtracted from the total acceleration vector, giving the true inertial acceleration at each point in time. Gravity vectors at each time point were saved for use in the PCA alignment algorithm.

The trimming of each smartphone’s data was performed in two parts for each pose (Figure 5). The manual selection of data was performed in MATLAB using the “ginput” function to graphically select three landmarks in the time series data to receive two time series data sets—one during the calibration maneuver and the other during the balance pose [16]. First, data from the calibration maneuver section of the experiment were manually trimmed for analysis. The start and end of this trimmed section was indicated by a large oscillation seen towards the beginning of the angular velocity data in the X direction. Secondly, the beginning of the participant’s jump at the end of the balance pose was manually selected. This was indicated by a large spike in angular velocity data towards the end of the data capture. From there, data were trimmed to start 3 s after the end of the calibration maneuver and end 3 s before the jump to give 24 total seconds of balance pose data. This trimming of data ensured the correct selection of each participant’s forward-flexion maneuver and balance pose for each smartphone.

PCA was performed on the handheld smartphone angular velocity data using the same functional alignment algorithm explained in detail in our recently published study [1]. In this algorithm, the angular velocity data trimmed during the forward-flexion calibration maneuver were used in PCA to identify a target vector representing the participant’s ML axis. Additionally, gravity vectors during the trimmed 24 s balance pose period were averaged over that time to obtain a mean gravity vector. The opposite of this gravity vector was taken to be the vertical target vector, representing the participant’s SI axis. Finally, the cross-product of the ML and SI target vectors was taken to produce a vector representing the participant’s AP axis. After the participant’s anatomical axes were determined, the handheld smartphone’s inertial acceleration and angular velocity data from both trimmed sections were projected to the anatomical AP, ML, and SI target vectors. 

Both the body strap and handheld smartphone’s trimmed angular velocity and acceleration data were filtered using a 3.5 Hz cutoff, zero-phase, 4th-order Butterworth filter.

For each smartphone and pose, balance metrics were produced by taking the RMS of acceleration data during the trimmed 24 s balance pose. RMS angular velocity scores were also produced during the calibration maneuver for each smartphone and pose, to be used in correlations (see Section 2.4). RMS angular velocity was used in correlations due to the nature of the forward-flexion maneuver being a rotational movement.

### 2.4. Statistical Methods

The statistical analysis of the study was conducted to determine (1) if there was a statistically significant correlation between the handheld vs. body strap smartphone’s angular velocity-based balance scores during the calibration maneuver, and (2) if each smartphone’s acceleration-based balance scores can detect within-participant differences between pose types. 

Pearson or Spearman’s rank correlation coefficients were used to address the first objective. Correlations were performed independently for each of three direction pairs on the smartphones. Smartphone directions for correlations were paired as follows: handheld AP and body strap Z direction, handheld ML and body strap X direction, and handheld SI and body strap Y direction. Individual groups were tested for normality using the Shapiro–Wilks test, with a significance level of 0.05 (α = 0.05). Any pairs with groups exhibiting non-normal data distributions were evaluated using Spearman’s rank correlation tests. All other groups were tested for correlation coefficients and significance using Pearson correlation tests. 

Kruskal–Wallis tests were used to investigate the second objective. Each direction (i.e., X, Y, and Z for the body strap smartphone, and AP, ML, and SI for the handheld smartphone) was analyzed separately for the detection of variance between within-participant pose types. Individual pose conditions for each direction (e.g., AP RMS acceleration for the T pose) were tested for normality using the Shapiro–Wilks test, with a significance level of 0.05 (α = 0.05). All groups contained data that violated normality. Kruskal–Wallis tests were performed independently for each smartphone (handheld and body strap), and post-hoc Dunn’s pairwise tests were performed on statistically significant (*p* < 0.05) groups. For post-hoc Dunn’s tests, an adjusted *p*-value of 0.0167, per a Bonferroni correction of 3 for multiple comparisons, was used to indicate statistically significant differences in means.

## 3. Results

### 3.1. Handheld vs. Body Strap Smartphone Correlation Results

The Pearson or Spearman correlation results between handheld and body strap smartphone RMS angular velocity scores derived during each participant’s calibration maneuvers are summarized in Table 1. The handheld AP and body strap Z direction RMS angular velocity scores violated normality using Shapiro–Wilks (*p* < 0.05). Given that result, Spearman correlations were used on the AP-Z group, and Pearson was used on the ML-X and SI-Y groups. Across 22 participants, the three balance poses produced 66 unique scores in each anatomical direction.

#### 3.1.1. AP Correlation Results

The results of the AP-Z correlation study are illustrated in Figure 6. The results show a moderate linear correlation for the body strap Z vs. handheld AP angular velocity scores (r = 0.491, *p* < 0.001). 

#### 3.1.2. ML Correlation Results

The results of the ML-X correlation study are illustrated in Figure 7. The results show a strong linear correlation for the body strap X vs. handheld ML angular velocity scores (r = 0.983, *p* < 0.001). 

#### 3.1.3. SI Correlation Results

The results of the SI-Y correlation study are illustrated in Figure 8. The results show a moderate linear correlation for the body strap Y vs. handheld ML angular velocity scores (r = 0.478, *p* < 0.001). 

### 3.2. Kruskal–Wallis Results

All RMS acceleration pose conditions for both smartphones and each direction were evaluated for normality using the Shapiro–Wilks test. All direction groups contained one or more pose conditions that violated normality (*p* < 0.05). For this reason, only Kruskal–Wallis tests were utilized to evaluate within-participant differences in pose difficulty for each direction using RMS acceleration scores, as summarized in Table 2. Both the handheld and body strap smartphones exhibited significant variance between pose types (*p* < 0.001) in the AP, ML, and SI directions. Post-hoc Dunn’s pairwise comparisons were performed for both the handheld and body strap smartphone data between all pose pairs. 

### 3.3. Pairwise Comparisons

For each smartphone, post-hoc Dunn’s pairwise comparisons of means were performed for all groups, given the statistical significance in Kruskal–Wallis results (*p* < 0.001). 

#### 3.3.1. AP Direction Pairwise Comparisons

The raw means and standard deviations for RMS acceleration scores are summarized in Table 3. 

Pairwise comparisons of the balance pose AP RMS acceleration scores are summarized for each smartphone (handheld and body strap) in Table 4. For the handheld smartphone, the SL vs. DL (*p* < 0.001) and T vs. DL (*p* = 0.0056) comparisons were found to be significant. For the body strap smartphone, only the SL vs. DL (*p* < 0.001) balance pose comparison demonstrated statistical significance, indicating that the T pose was not significantly different from the SL and DL poses.

#### 3.3.2. ML Direction Pairwise Comparisons

Raw means and standard deviations for RMS acceleration scores are summarized in Table 5. 

Pairwise comparisons of balance pose ML RMS acceleration scores are summarized for each smartphone (handheld and body strap) in Table 6. For the handheld and body strap smartphones, the SL vs. DL (*p* < 0.001) and T vs. DL (*p* = 0.0013, *p* = 0.0005) comparisons demonstrate statistical significance.

#### 3.3.3. SI Direction Pairwise Comparisons

Raw means and standard deviations for RMS acceleration scores are summarized in Table 7. 

Pairwise comparisons of balance pose SI RMS acceleration scores are summarized for each smartphone (handheld and body strap) in Table 8. For the handheld smartphone, significant differences were demonstrated for both the SL vs. DL (*p* < 0.001) and the T vs. SL (*p* = 0.0134) pose pairs. For the body strap smartphone, only the SL vs. DL (*p* < 0.001) balance pose comparison was proven to be statistically significance, indicating that the T pose was not significantly different from the SL and DL poses.

### 3.4. Bar Graph Comparisons

The handheld and body strap smartphone results for each pose type are visualized in Figure 9 and Figure 10, respectively.

### 3.5. Calibration Algorithm Validation

The PCA functional alignment algorithm was able to successfully project angular velocity data from the handheld smartphone onto a global anatomical coordinate system (i.e., the AP, ML, and SI axes). Note that since the forward-flexion calibration maneuver is a one-dimensional rotational maneuver about the participant’s ML axis, most of the angular velocity component occurs about the ML axis, which is seen in the graph on the right of Figure 11 below. 

## 4. Discussion

### 4.1. Summary of Key Findings

The handheld smartphone with functional alignment successfully produced balance scores that were moderately to strongly correlated with the body strap smartphone with assumed alignment. The moderate to strong correlation in RMS angular velocity scores during the calibration maneuver (r = 0.487–0.983, *p* < 0.001) indicate that the PCA functional alignment effectively aligned the handheld smartphone’s axes with the global anatomical axes. The handheld and body strap smartphones were most strongly correlated in the AP and ML directions. 

Additionally, both smartphones successfully detected variations between poses in within-participant RMS acceleration scores in each anatomical direction. Notably, pairwise comparisons show that the handheld smartphone successfully detected significant differences between two pose type pairs for the AP, ML, and SI directions. The body strap smartphone only detected significant differences between two pose-type pairs in the ML direction. The smartphones performed comparatively well in detecting significant differences in pose type in the ML direction, but the handheld smartphone performed better in the AP and SI directions. In summary, the handheld phone was able to detect significant differences in six pose-type pairs, whereas the body strap smartphone only detected significant differences in four pose-type pairs. 

Note that RMS acceleration scores were nominally higher for the handheld smartphone (see Figure 9 and Figure 10). This could be due to the handheld smartphone’s placement being superior relative to the body strap phone. This would result in greater acceleration due to its greater distance to the participant’s center of sway.

Overall, these results indicate that the handheld smartphone with functional alignment performed better than the body strap smartphone with assumed alignment in assessing changes in participant balance ability. While assumed alignment using a body strap is still a valid and effective method for assessing static balance ability, PCA functional alignment could eliminate the need for purchasing and using a well-fitting body strap. 

This study’s findings were consistent with our group’s previous results in several respects. First, we were able to confirm that the PCA algorithm for post-processing balance data—which we validated for use with a smartwatch in [1]—was effective for use with a smartphone. Second, we found that the smartphone with PCA calibration approach used in this study was able to distinguish between different balance poses in the AP, ML, and SI directions, while the smartwatch with PCA approach in [1] only did so in the ML direction, and not AP (SI was not evaluated). Our previous smartphone study without PCA calibration [17] was only able to distinguish between poses in the ML (not AP) direction, suggesting that our PCA algorithm helped improve the resolution of our outcome metrics in the AP direction.

While other research groups have successfully used IMUs to measure clinically relevant balance metrics, the majority have focused on standalone IMUs rather than those embedded in smartphones and smartwatches [12]. The use of commonly owned devices makes the analyses more translatable for everyday health monitoring. More and more studies are using smartphones to measure balance with moderate to high validity and reliability [6,10,13,18], but these studies have relied on a body strap to manually align the sensors with the body axes, which is prone to errors. Our use of PCA to align the smartphone’s axes with the anatomical axes is novel and shows great promise in making balance assessments more accessible to the everyday smartphone user.

### 4.2. Study Limitations

The limitations of this study include its relatively low sample size, relatively narrow age range, and lack of validation with force plate data—which is considered the gold standard method for balance assessment. Other balance parameters, such as RMS angular velocity during the balance poses, were not analyzed. Additionally, the forward-flexion maneuver may not be the most accessible maneuver for those with lower-back injuries or other muscular injuries or neurological diseases. The target population, which includes individuals with neurological conditions such as Parkinson’s disease, may face challenges due to symptoms such as hand tremors and significant balance difficulties. These symptoms can make it difficult to hold an iPhone steadily against the chest and to maintain various balance poses, potentially impacting the effectiveness of the balance assessment method—a solution could be to strap the iPhone to the user’s hand. 

One of the next steps is to increase the sample size to better reflect the population and provide insight related to the objectives of this study. Additionally, further investigations into more accessible calibration maneuvers may yield similar results to those found in this study. 

### 4.3. Future Development of Accessible Balance Assessment 

State-of-the art IMUs found in smartphones have the potential to allow for accessible and accurate balance assessments in at-home and other real-world settings. The results of this study and our previous smartwatch study [1] indicate that IMU-enabled devices may be held in a comfortable position and successfully record significant changes in balance ability using functional calibration. The PCA calibration method may provide an increasingly accurate and accessible method for tracking balance ability, especially for those with limited access to physicians or care facilities specializing in this assessment.

With smartphones becoming ever more prevalent in people’s daily lives, they have the potential to democratize access to care in times and situations where in-person care may not be available. A recent example of this was seen with the worldwide COVID-19 pandemic, in which the United States saw reductions in access to care in patients with chronic conditions [19]. This reduction in access to care occurred disproportionally with annual household income, race, and self-reported health status [18]. Critical research gaps exist related to the feasibility and measurement methods of remote assessments [20]. Some include logistical issues, such as the environment in which users would perform remote balance assessments. With the emergence of data-driven methods, enabled by artificial intelligence and machine learning, data collected during remote assessments may have the potential to be used both in the diagnoses of balance-related disorders and in patient rehabilitation. Large and diverse participant population sets must be sampled, and extensive validation research must be performed. With large-scale cross-functional collaborative research in laboratories already involved in wearable-based balance research, this could become realized. Combining data-driven methods with both physics-based and data pre-processing methods, such as PCA, has the potential to revolutionize balance assessment. 

## Figures and Tables

**Figure 1 sensors-24-05467-f001:**
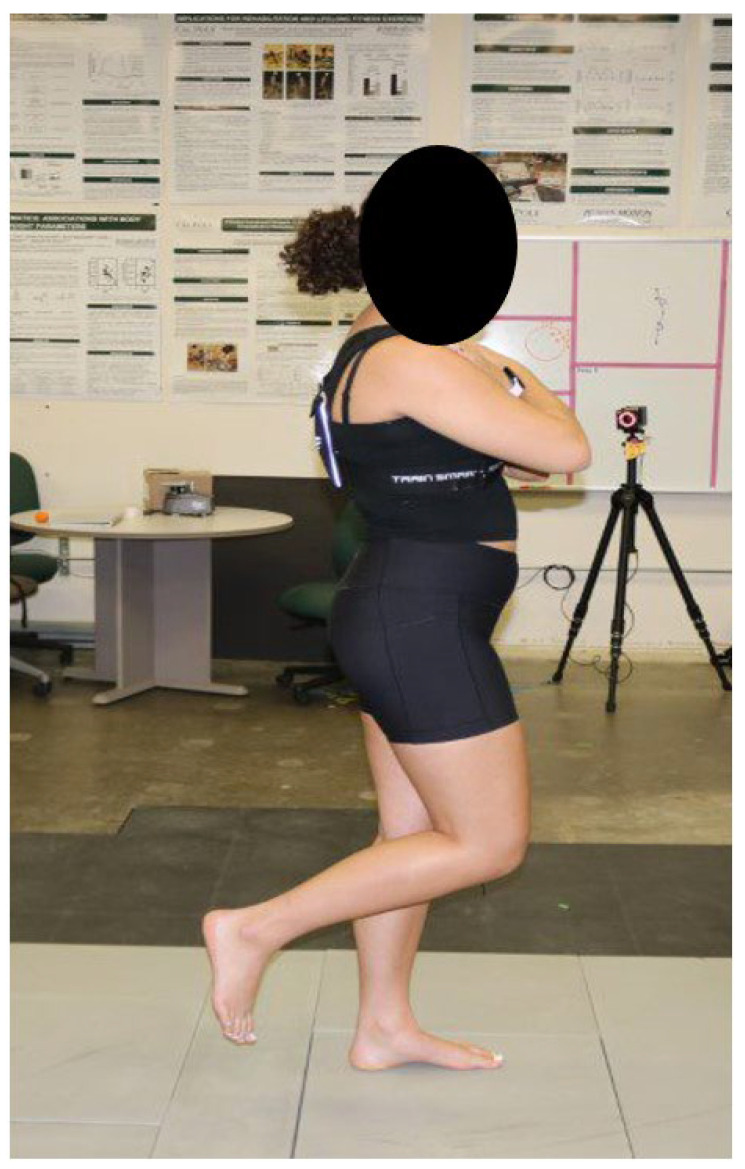
Participant performing balance pose during experiment while wearing a smartphone in a body strap with assumed alignment with the participant’s anatomical axes. Smartphone was aligned on the participant’s back with the phone’s Z-axis facing in the participant’s anterior direction and Y-axis in the participant’s superior direction.

**Figure 2 sensors-24-05467-f002:**
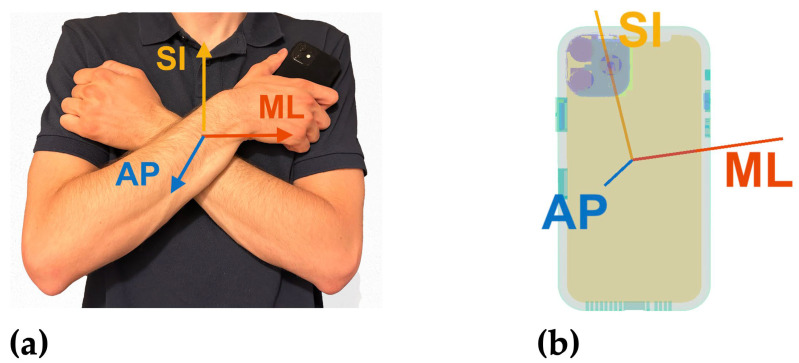
(**a**) Handheld smartphone held against chest, with participant’s anatomical axes displayed. (**b**) Overlay of handheld smartphone with calculated anteroposterior (AP, shown in blue), mediolateral (ML, shown in red), and superoinferior (SI, shown in yellow) axes after calibration maneuver PCA analysis.

**Figure 3 sensors-24-05467-f003:**
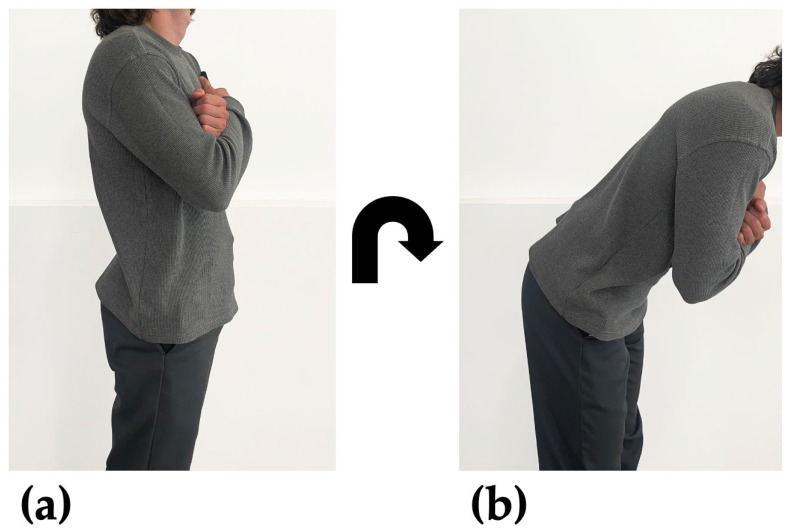
Participants performed the forward flexion maneuver before every balance pose with handheld and body strap smartphones (not pictured) recording inertial data. (**a**) Participant standing in upright position. (**b**) Participant bent about their ML axis in forward-flexion. The forward-flexion calibration maneuver consists of moving from position (**a**) to (**b**) and back to (**a**).

**Figure 4 sensors-24-05467-f004:**
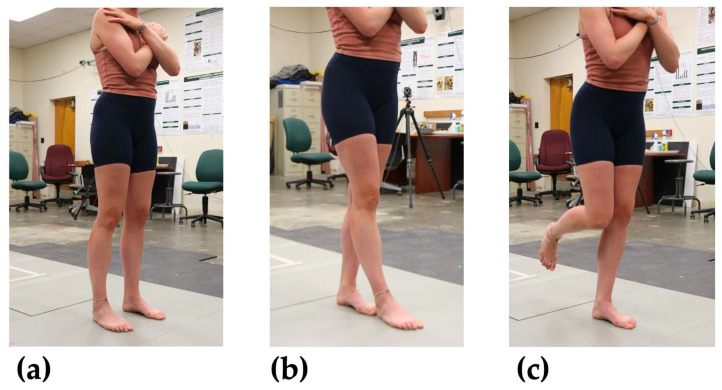
Images of three balance poses of increasing difficulty. (**a**) Double-leg stance (DL); expected to be least difficult. (**b**) Tandem stance (T); expected to be of moderate difficulty. (**c**) Single-leg stance (SL); expected to be most difficult.

**Figure 5 sensors-24-05467-f005:**
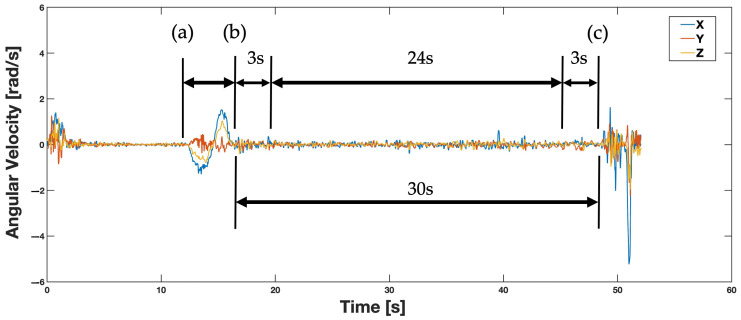
Smartphone manual data trimming. (a) Beginning of calibration maneuver. (b) End of calibration maneuver. (c) Beginning of jump, end of balance pose. Also shown, 30 s balance pose between points (b) and (c), and trimming to 24 s period starting 3 s after (b) and before (c). Figure key shows color coding for X, Y, and Z axis inertial data.

**Figure 6 sensors-24-05467-f006:**
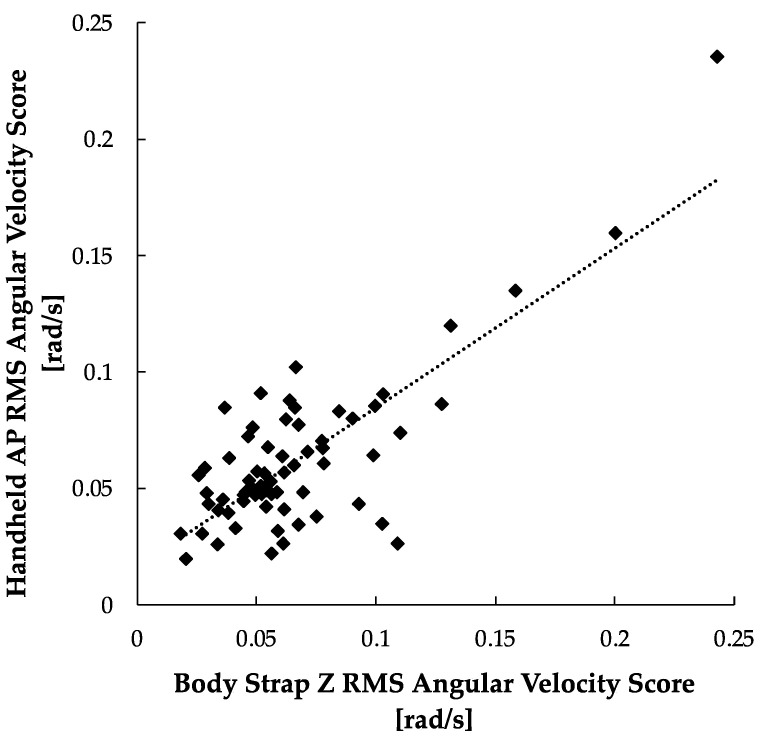
22 AP-Z RMS angular velocity scores between handheld and body strap smartphone. A positive trendline was produced (r = 0.491, *p* < 0.001).

**Figure 7 sensors-24-05467-f007:**
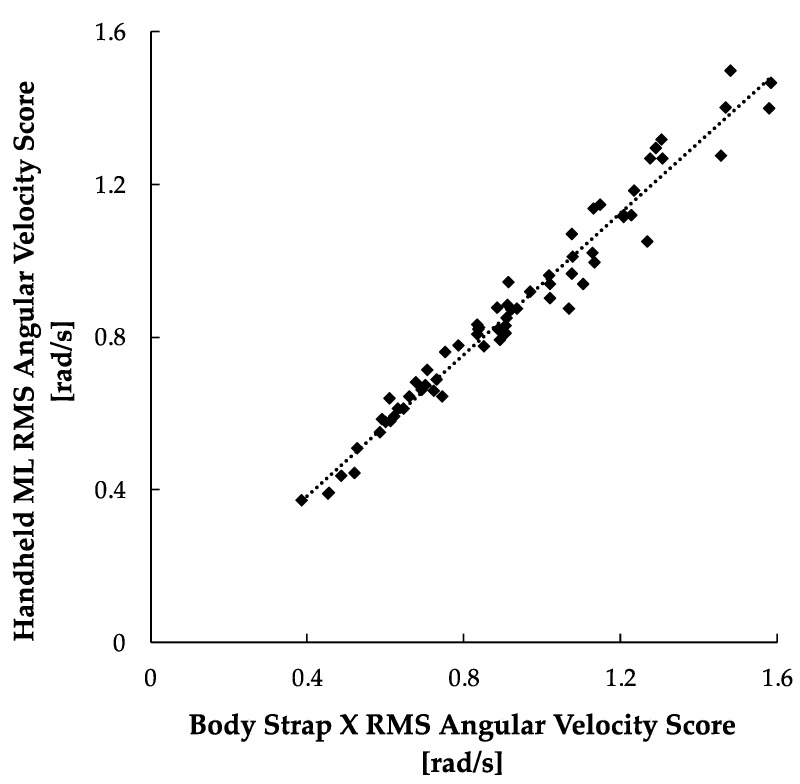
22 ML-X RMS angular velocity scores between handheld and body strap smartphone. A positive trendline was produced with the calibration method (r = 0.983, *p* < 0.001).

**Figure 8 sensors-24-05467-f008:**
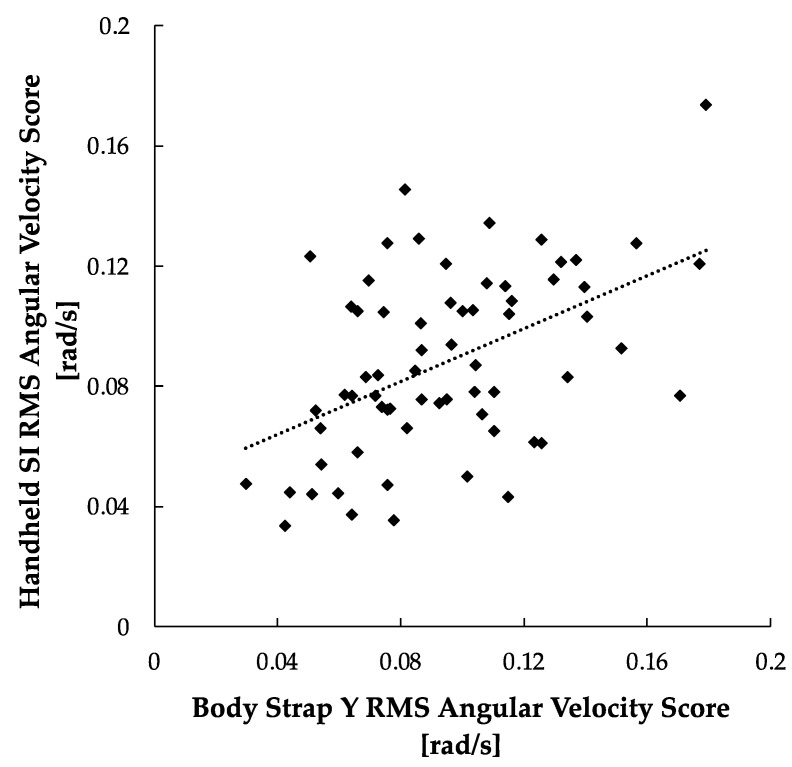
22 SI-Y RMS angular velocity scores between handheld and body strap smartphone. A positive trendline was produced with the calibration method (r = 0.487, *p* < 0.001).

**Figure 9 sensors-24-05467-f009:**
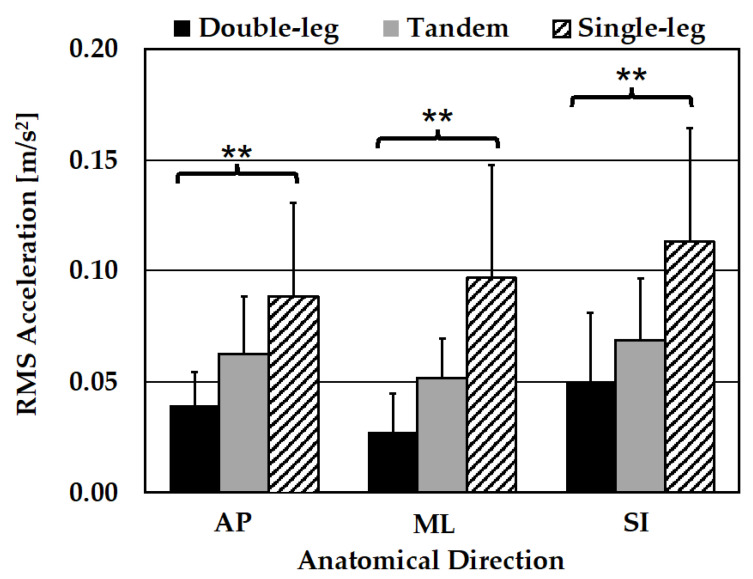
RMS acceleration scores in each direction and for each balance pose for the handheld smartphone; mean and 1 standard deviation shown. ** = significant differences (*p* < 0.0167) between two or more balance pose pairs in post-hoc Dunn’s pairwise comparisons.

**Figure 10 sensors-24-05467-f010:**
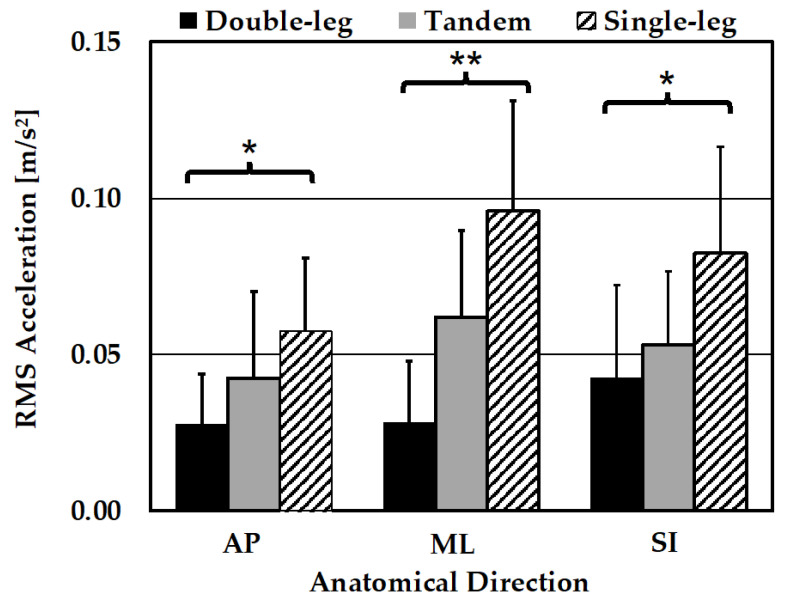
RMS acceleration scores in each direction and for each balance pose for the body strap smartphone; mean and 1 standard deviation shown. * = significant differences (*p* < 0.001) between all 3 balance poses in Kruskal–Wallis test. ** = significant differences (*p* < 0.0167) between two or more balance pose pairs in post-hoc Dunn’s pairwise comparisons.

**Figure 11 sensors-24-05467-f011:**
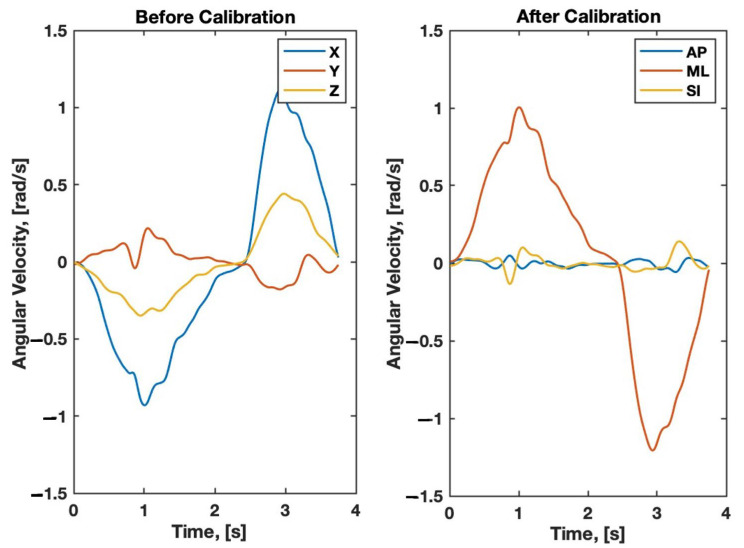
Sample of angular velocity data during the forward-flexion maneuver before (**left**) and after (**right**) PCA functional alignment of the handheld smartphone. (**Left**) Filtered angular velocity data from the handheld smartphone in the inertial X, Y, and Z axes prior to PCA calibration. (**Right**) Data in the AP, ML, and SI axes after PCA calibration.

**Table 1 sensors-24-05467-t001:** Correlation coefficients and significance of RMS angular velocity scores between handheld and body strap smartphones. Correlation direction pairs were selected based on expectation of directional alignment (e.g., handheld ML axis with body strap X axis).

	AP-Z **	ML-X *	SI-Y *
Correlation coefficient	0.491	0.983	0.487
*p*-value	<0.001	<0.001	<0.001
N	66	66	66

* Indicates Pearson correlation was used. ** Indicates Spearman’s rank correlation was used.

**Table 2 sensors-24-05467-t002:** Kruskal–Wallis results for acceleration-based balance scores. Degrees of freedom (DOF), χ², and *p*-values are displayed to represent within-subjects pose effects on RMS score.

Device	Direction	DOF	χ²	*p*-Value
Handheld with functional alignment	AP	2	27.387	<0.001 *
	ML	2	39.662	<0.001 *
	SI	2	22.737	<0.001 *
Body strap with assumed alignment	AP	2	24.123	<0.001 *
	ML	2	41.897	<0.001 *
	SI	2	19.477	<0.001 *

* Indicates statistical significance.

**Table 3 sensors-24-05467-t003:** AP RMS acceleration score means (M) and standard deviations (SD) for each pose and for both handheld and body strap smartphones.

Device	Pose	M [m/s^2^]	SD [m/s^2^]
Handheld with functional alignment	DL	0.0396	0.0148
	T	0.0627	0.0258
	SL	0.0884	0.0419
Body strap with assumed alignment	DL	0.0278	0.0159
	T	0.0424	0.0280
	SL	0.0575	0.0233

**Table 4 sensors-24-05467-t004:** Dunn’s pairwise comparison results for acceleration-based balance scores in the AP direction.

Device	Pose Comparison	Z	*p*-Value
Handheld with functional alignment	SL vs. DL	5.191	<0.001 *
	T vs. DL	3.110	0.0056 *
	T vs. SL	−2.073	0.1144
Body strap with assumed alignment	SL vs. DL	4.900	<0.001 *
	T vs. DL	2.599	0.0280
	T vs. SL	−2.293	0.0655

* Indicates statistical significance (*p* < 0.0167).

**Table 5 sensors-24-05467-t005:** ML RMS acceleration score means (M) and standard deviations (SD) for each pose and for both handheld and body strap smartphones.

Device	Pose	M [m/s^2^]	SD [m/s^2^]
Handheld with functional alignment	DL	0.0275	0.0171
	T	0.0518	0.0178
	SL	0.0968	0.0506
Body strap with assumed alignment	DL	0.0286	0.0191
	T	0.0621	0.0277
	SL	0.0958	0.0353

**Table 6 sensors-24-05467-t006:** Dunn’s pairwise comparison results for acceleration-based balance scores in the ML direction.

Device	Pose Comparison	Z	*p*-Value
Handheld with functional alignment	SL vs. DL	6.275	<0.001 *
	T vs. DL	3.510	0.0013 *
	T vs. SL	−2.757	0.0175
Body strap with assumed alignment	SL vs. DL	6.432	<0.001 *
	T vs. DL	3.777	0.0005 *
	T vs. SL	−2.647	0.0244

* Indicates statistical significance (*p* < 0.0167).

**Table 7 sensors-24-05467-t007:** SI RMS acceleration score means (M) and standard deviations (SD) for each pose and for both handheld and body strap smartphones.

Device	Pose	M [m/s^2^]	SD [m/s^2^]
Handheld with functional alignment	DL	0.0504	0.0307
	T	0.0688	0.0278
	SL	0.1129	0.0511
Body strap with assumed alignment	DL	0.0428	0.0293
	T	0.0533	0.0234
	SL	0.0825	0.0338

**Table 8 sensors-24-05467-t008:** Dunn’s pairwise comparison results for acceleration-based balance scores in the SI direction.

Device	Pose Comparison	Z	*p*-Value
Handheld with functional alignment	SL vs. DL	4.728	<0.001 *
	T vs. DL	1.877	0.1816
	T vs. SL	−2.843	0.0134 *
Body strap with assumed alignment	SL vs. DL	4.366	<0.001 *
	T vs. DL	1.673	0.2831
	T vs. SL	−2.686	0.0217

* Indicates statistical significance (*p* < 0.0167).

## Data Availability

Please direct data inquiries to Britta Berg-Johansen, at bbergjoh@calpoly.edu. Some raw data may be unavailable due to participant confidentiality agreements.

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
