# Peer review of "Balance Assessment Using a Handheld Smartphone with Principal Component Analysis for Anatomical Calibration"

_sensors, 2024, doi:10.3390/s24175467_

Round 1

Reviewer 1 Report

Comments and Suggestions for Authors

Thank you for inviting to me review this manuscript.

Research into analysing aspects of human movement by ‘wearables’ instead of a camera-based system and force plates in motion analysis lab is becoming increasingly common. As the authors describe, wearables that are relatively cheap, are easy to fit by patients or their carers and provide data, which is meaningful as well as reliable, valid and easy to interpret by clinicians would have numerous applications.

This manuscript describes how data captured by hand-held mobile phone compare to the data captured by a ‘strictly placed’ mobile phone.

I am afraid I have several issues with the study design and data analysis. The authors describe that the disadvantages of the ‘strictly’ placed mobile phone to measure balance include that this requires the phone to be fixed to the trunk with a body strap and assumes alignment of the smartphone axes with the anatomical body axes. However, the authors did not describe how the ‘strictly placed mobile phone’ was aligned with anatomical landmarks.  Further,  why did the authors choose to apply to the PCA algorithm for calibration through a flexion manoeuvre to a data captured by a hand held mobile phone and not one positioned in a body harness (without attempting it to align with the anatomical axes)?  In other words, why is it better to hold a phone than to position it an easy to use, comfortable harness/strap?

Looking at the data in Figure 7 and 8, the accelerations of the handheld phone are somewhat higher than the ‘strictly placed’ phone, suggesting that there may have been some additional movement in hands/arm holding the phone in addition to movement of the trunk. In fact, this may explain why the handheld phone was able to distinguish between more poses as participants may have slightly moved the hand/arm holding the phone during some of the more challenging poses.

In some clinical populations, holding a phone may not be possible and/or significantly influence the balance performance.

Further, why did the authors choose to compare the RMS of the angular velocity date between the phones and not the acceleration data during the three different poses? Also, the angular velocity data was recorded during an actual movement (forward flexion), rather than a 30 s ‘static pose’ so is it appropriate to use the RMS of the angular velocity for the analysis?

I would recommend the authors to review how they report the statistical tests. I have highlighted several examples in the comments in the pdf.

The discussion is very brief. How do the results compare to those by other studies?

See for specific comments the comments to the pdf.

Author Response

Thank you very much for taking the time to review this manuscript. Please find the detailed responses below and the corresponding revisions as highlighted in the re-submitted files.

Comment 1: The authors describe that the disadvantages of the ‘strictly’ placed mobile phone to measure balance include that this requires the phone to be fixed to the trunk with a body strap and assumes alignment of the smartphone axes with the anatomical body axes. However, the authors did not describe how the ‘strictly placed mobile phone’ was aligned with anatomical landmarks.

Response 1: Thank you for pointing this out. We agree with this comment. Therefore, we have addressed this comment by providing more information in section 2.2 about the placement of the strictly placed smartphone. Please see lines 118-121 in the paper. Additionally, we also included a figure showing a photo of a participant wearing a chest strap while performing a balance pose (Figure 1).

Lines 118-121:

“Two sizes of chest straps – one larger and one smaller – were utilized to accommodate participants of varying size. Anatomical placement of the strictly placed smartphone was not standardized due to a lack of adjustability of the chest straps.”

Comment 2: Why did the authors choose to apply to the PCA algorithm for calibration through a flexion maneuver to a data captured by a handheld mobile phone and not one positioned in a body harness (without attempting it to align with the anatomical axes)?  In other words, why is it better to hold a phone than to position it an easy to use, comfortable harness/strap?

Response 2: Thank you for pointing this out. We want to clarify that the reasoning for using the PCA alignment algorithm is to allow for the smartphone to be held in the comfortable handheld position. We were interested in developing a way to evaluate balance in a handheld position because it would eliminate the need for someone to purchase a chest strap, therefore potentially increasing the accessibility of balance assessment with a smartphone. Please see lines 79-84 in the manuscript for clarification on this.

Lines 79-84:

“The precise placement of these sensors is often challenging due to the irregular shape of the human limbs and anatomical variability from person to person, as well as the size constraints and adjustability of body straps. Additionally, with strict placement possibly requiring the purchase of a well-fitting body strap, this may further reduce the accessibility of this balance assessment method.”

Comment 3: Looking at the data in Figure 7 and 8, the accelerations of the handheld phone are somewhat higher than the ‘strictly placed’ phone, suggesting that there may have been some additional movement in hands/arm holding the phone in addition to movement of the trunk. In fact, this may explain why the handheld phone was able to distinguish between more poses as participants may have slightly moved the hand/arm holding the phone during some of the more challenging poses.

Response 3: Thank you for pointing this out. We agree with this comment. The acceleration scores for the handheld smartphone were nominally higher than the strictly placed phone. We believe this could be because the handheld smartphone was placed higher relative to the participant’s body (superiorly). This would result in the accelerations seen by that phone to be higher than the strictly placed phone’s since it is further from the center of rotation of the participant. We have clarified this in lines 381-384 in the discussion section (section 4.1) of the manuscript. Please note that since two additional figures have been added, Figures 7 and 8 are now Figures 9 and 10.

Lines 381-384:

“Note that RMS acceleration scores were nominally larger for the handheld smartphone. This could be due to the handheld smartphone’s placement in the superior direction relative to the strictly placed phone. This would result in larger accelerations due to its greater distance to the participant’s center of sway.”

Comment 4: In some clinical populations, holding a phone may not be possible and/or significantly influence the balance performance.

Response 4: Thank you for pointing this out. We agree with this comment. We understand that those with neurological disorders, such as Parkinson’s disease, may have a difficult time holding a phone, which may also result in poor performance of this balance assessment method. This has been addressed in lines 415-420 in the discussion section of the manuscript (section 4.3).

Lines 415-420:

The target population, which includes individuals with neurological conditions like Parkinson’s disease, may face challenges due to symptoms such as hand tremors and significant balance difficulties. These symptoms can make it difficult to hold an iPhone steadily against the chest and to maintain various balance poses, potentially impacting the effectiveness of the balance assessment method.”

Comment 5: Why did the authors choose to compare the RMS of the angular velocity data between the phones and not the acceleration data during the three different poses? Also, the angular velocity data was recorded during an actual movement (forward flexion), rather than a 30 s ‘static pose’ so is it appropriate to use the RMS of the angular velocity for the analysis?

Response 5: We want to clarify that RMS of acceleration during the 30s balance poses was compared between the phones in the Kruskal-Wallis analyses, not RMS angular velocity. RMS angular velocity was recorded during the forward flexion maneuver and compared between the phones in the correlation tests. Correlation tests were used with RMS angular velocity during the calibration maneuver to determine the strength of correlation between the two phones.

Comment 6: I would recommend the authors to review how they report the statistical tests. I have highlighted several examples in the comments in the pdf.

Response 6: We want to thank you for the comments you left in regard to our statistical analysis of our data. We have adjusted our statistical analyses extensively to address the comments. We performed Shapiro-Wilks tests for normality on all RMS angular velocity data used in correlation tests and RMS acceleration data used in analysis of variance tests. In correlation tests, we found that our AP-Z group did not exhibit normality (p<0.05), which warranted the use of Spearman’s rank correlation test for that group. All other correlation tests continued to use Pearson correlation tests. Please see lines 225-233 (section 2.4) in the manuscript.

Lines 225-233:

“Pearson or Spearman’s rank correlation coefficients were used to address the first objective. Correlations were performed independently for each of three direction pairs on the smartphones. Directions for correlations were paired as follows: handheld AP and strictly placed Z direction, handheld ML and strictly placed X direction, and handheld SI and strictly placed Y direction. Individual groups were tested for normality using the Shapiro-Wilks test, with a significance level of 0.05 (α = 0.05). Any pairs with groups exhibiting non-normal data distributions were evaluated using Spearman’s rank correlation tests. All other groups were tested for correlation coefficients and significance using Pearson correlation tests.”

For the analysis of variance tests, we found that all groups contained RMS acceleration data (one or more of the three pose data groups in each smartphone direction group) that was not normal (p<0.05). To address this, we used Kruskal-Wallis tests to assess differences between poses for each direction. We also used post-hoc Dunn’s pairwise comparisons to assess differences between individual pose pairs, with an adjusted p-score of 0.0167 for a Bonferroni correction of 3 (0.05/3), as was suggested by you in the pdf comments. Please see lines 234-244 (section 2.4).

Lines 234-244:

“Kruskal-Wallis tests were used to investigate the second objective. Each direction (i.e., X, Y, and Z for the strictly placed smartphone, and AP, ML, and SI for the handheld smartphone) was analyzed separately for detection of variance between within-participant pose types. Individual pose groups for each direction (e.g., AP RMS acceleration for the T pose) were tested for normality using the Shapiro-Wilks test, with a significance level of 0.05 (α = 0.05). All groups contained data that violated normality. Kruskal-Wallis tests were performed independently for each smartphone (handheld and strictly placed), and post-hoc Dunn’s pairwise tests were performed on statistically significant (p<0.05) groups. For post-hoc Dunn’s tests, an adjusted p-value of 0.0167, per a Bonferroni correction of 3 for multiple comparisons, was used to indicate statistically significant differences in means.”

Comment 7: The discussion is very brief. How do the results compare to those by other studies?

Response 7: We have expanded our discussion substantially. We have included two paragraphs on comparisons to other studies, including some of our previous studies within our lab. We summarize that in our study, we were able to determine significant differences between balance poses in both the AP and ML directions, whereas our previous smartwatch study was only able to in the ML direction. Please see lines 388-407 in the manuscript for additional commentary.

Lines 388-407:

“This study’s findings were consistent with our group’s previous results in several respects. First, we were able to confirm that the PCA algorithm for post-processing balance data – which we validated for use with a smartwatch in [1] – was effective for use with a smartphone. Second, we found that the smartphone with PCA calibration approach used in this study was able to distinguish between different balance poses in the AP, ML, and SI directions, while the smartwatch with PCA approach in [1] only did in the ML direction, not AP (SI was not evaluated). Our previous smartphone study without PCA calibration [17] was only able to distinguish between poses in the ML (not AP) direction, suggesting that our PCA algorithm helped improve resolution of our outcome metrics in the AP direction.

While other research groups have successfully used IMUs to measure clinically relevant balance metrics, the majority have focused on standalone IMUs rather than those embedded in smartphones and smartwatches [12]. The use of commonly owned devices makes the analyses more translatable for everyday health monitoring. More and more studies are using smartphones to measure balance with moderate to high validity and reliability [6, 10, 13, 18], but these studies have relied on strict placement of the smartphone to manually align the sensors with the body axes, which is prone to errors and requires the use of a belt or strap. Our use of PCA to align the smartphone’s axes with the anatomical axes is novel and shows great promise to make balance assessment more accessible to the everyday smartphone user.”

Additional Clarifications: We also wanted to add that we included a new figure (Figure 3) that shows how the forward flexion maneuver would be performed during experiment. We also wanted to address that foot positioning was not standardized in any of the balance poses. We did add in the methods section (line 151) that participants were asked to stand shoulder width apart during the double leg stance (DL).

Reviewer 2 Report

Comments and Suggestions for Authors

In this paper, the authors demonstrate the use of a cell phone for balance assessment studies. This work is based on a clever idea, where a smartphone is used to determine different balance poses.  Despite the fact that P-value indicates indeed a statistical significance, I think a larger sample would provide better insights for the hypothesis of this work. I am curious to see if this methodology could applied to population group like, Parkinson-condition patients, as it might be quite challenging. In any case, this is indeed a solid research paper which could add one more tool to the therapists for monitoring cost-less patients.

Author Response

Thank you very much for taking the time to review this manuscript. We greatly appreciate your positive feedback. Please find responses to your two comments below and the corresponding revisions as highlighted in the re-submitted files.

Comment 1: Despite the fact that P-value indicates indeed a statistical significance, I think a larger sample would provide better insights for the hypothesis of this work.

Response 1: We completely agree with this statement as a larger sample size almost always provides more insight into a hypothesis. We do state the smaller sample size as a limiting factor in the Study Limitations but have now added an additional sentence to elaborate on next steps. This can be found in lines 421-423.

Lines 421-423:

“One of the next steps is to increase the sample size to better reflect the population and provide better insight on the objectives of this study.”

Comment 2: I am curious to see if this methodology could applied to population group like, Parkinson-condition patients, as it might be quite challenging.

Response 2: It could be challenging due to Parkinson-condition patients since balance issues are a quite common symptom and the iPhone might be challenging to hold steadily. One solution could be to strap the iPhone to the user’s hand to ensure a more stable position. This is a population that we would like our work to target in future studies. Further information has been added to the Discussion, as seen in lines 415-420.

Lines 415-420:

The target population, which includes individuals with neurological conditions like Parkinson’s disease, may face challenges due to symptoms such as hand tremors and significant balance difficulties. These symptoms can make it difficult to hold an iPhone steadily against the chest and to maintain various balance poses, potentially impacting the effectiveness of the balance assessment method.”

Reviewer 3 Report

Comments and Suggestions for Authors

The objectives of this study is to determine if correlations existed between angular velocity scores from a handheld smartphone with PCA alignment vs. a strictly placed smartphone and analyze acceleration score differences across balance poses of increasing difficulty. 

I understandr that the handheld and strictly placed smartphones were most strongly correlated in the AP and ML directions.
The results are enough to indicate that the handheld smartphone with PCA calibration performed better than the strictly placed smartphone at assessing changes in participant balance ability.

There are several points which need to be discussed.

Is the IMU sensor data from the iPhone processed and optimized by the iPhone?

Even if the iPhone is not fixed, isn't the information filtered in the same way as if it were fixed? 

These are the points I was wondering about. I think it would be better to clarify these points.

Author Response

Thank you very much for taking the time to review this manuscript. We greatly appreciate your positive feedback. Please find responses to your two comments below and the corresponding revisions as highlighted in the re-submitted files.

Comment 1: Is the IMU sensor data from the iPhone processed and optimized by the iPhone?

Response 1: Yes, the IMU sensor data from the iPhone is processed and optimized by the iPhone. Additional information was put into the methods section in lines 167-172.

Lines 167-172:

“The IMU sensor data from the iPhone is processed and optimized through sensor fusion and filtering. The iPhone is equipped with a LIS302DL 3-axis MEMS-based accelerometer that uses a low-pass filter to reduce high frequency noise and smooth out the data. The MATLAB mobile app uses additional sensor fusion algorithms to estimate orientation and position over time from the accelerometer, gyroscope, and magnetometer of the iPhone.”

Comment 2: Even if the iPhone is not fixed, isn't the information filtered in the same way as if it were fixed?

Response 2: Both iPhones are filtered using a 4th order Butterworth filter as stated in the Data Acquisition and Post Processing to achieve a smoother and more consistent output. This statement can be found in lines 205-207.

Lines 205-207:

“Both the strictly placed and handheld smartphone’s trimmed angular velocity and acceleration data were filtered using a 3.5 Hz cutoff, zero-phase, 4th order Butterworth filter.

Round 2

Reviewer 1 Report

Comments and Suggestions for Authors

thank you for your addressing my comments, I think the article is much improved.

However, I still have a some further comments.

I still think that referring to a ‘strictly positioned’ smartphone for the condition in which the smartphone was placed in a chest strap but without referring to anatomical landmarks is confusing.  The authors basically compared a smartphone positioned in a body strap compared to a handheld one.

For use in a  clinical population, I still can’t see why a simple strap cannot be used instead of holding the phone in one hand.

Referring to ‘pose groups’ instead of ‘pose conditions’ throughout the article may also lead to unclarity.

I am not sure why the authors added the figures 9 & 10, which although clear,  basically provide the same information as the tables

Is the accepted standard not p<0.001 (instead of p<0.0001?)

Abstract : an r of 0.487 is usually not defined as a ‘strong’ correlation

Author Response

Thank you so much for the additional comments on the manuscript. We have our responses below and have made changes accordingly.  

Comment 1: 

I still think that referring to a ‘strictly positioned’ smartphone for the condition in which the smartphone was placed in a chest strap but without referring to anatomical landmarks is confusing.  The authors basically compared a smartphone positioned in a body strap compared to a handheld one. 

Response 1: 

Thank you for this comment. We want to clarify that the smartphone that was placed in the chest strap was placed on the participant’s lower back. We used this smartphone in this position to assume alignment of the smartphone’s axes with the participant’s anatomical axes. We have adjusted the wording in our manuscript, changing “strictly placed” to “assumed alignment” to better reflect what we did. We have also included a sentence about other studies that have used assumed alignment on the lower back. Please see lines 79-81. 

Lines 79-81:  

“A recent review of studies using wearable sensors to evaluate standing balance found that 81% of  studies also placed the device on the lower back [14].” 

Comment 2: 

For use in a clinical population, I still can’t see why a simple strap cannot be used instead of holding the phone in one hand. 

Response 2: 

Thank you for your comment. We understand and agree that it is completely possible to use a strap instead of holding a phone in order to measure balance. The purpose of our study was to validate the use of a handheld device with functional alignment with a smartphone in a strap. We are doing this to make balance assessment more accessible for situations where the participant is outside of the clinic and may not have access to a strap of this kind. Please see lines 393-396 in the discussion.  

Lines 393-396: 

“While assumed alignment using a body strap is still a valid and effective method for assessing static balance ability, PCA functional alignment could eliminate the need for purchasing and using a well-fitting body strap.” 

Comment 3: 

Referring to ‘pose groups’ instead of ‘pose conditions’ throughout the article may also lead to unclarity. 

Response 3: 

Thank you for this comment, we agree and have changed this throughout the manuscript.  

Comment 4: 

I am not sure why the authors added the figures 9 & 10, which although clear,  basically provide the same information as the tables 

Response 4: 

Thank you for this comment. We believe that figures 9 and 10 add a clear visual representation of our data output. These figures were present in the original manuscript that was submitted, and we were asked to include raw data, which is why the new tables with means and standard deviations are now included. The editor also requested that we make the manuscript longer, and one way we decided to do that was to present both tables and figures for added clarity. 

Comment 5: 

Is the accepted standard not p<0.001 (instead of p<0.0001?) 

Response 5: 

Thank you for this comment. We agree, p<0.001 is a more accepted standard. This change has been implemented throughout the manuscript.  

Comment 6: 

Abstract : an r of 0.487 is usually not defined as a ‘strong’ correlation 

Response 6: 

Thank you for this comment. We agree, and have adjusted the language throughout our manuscript to reflect that our correlation strengths were moderate to strong. See an example of this in lines 265-267. 

Lines 265-267:  

“Results show a moderate linear correlation for the body strap Z vs. handheld AP angular velocity scores (r = 0.491, p<0.001).”